# Recent Developments of Liquid Chromatography Stationary Phases for Compound Separation: From Proteins to Small Organic Compounds

**DOI:** 10.3390/molecules27030907

**Published:** 2022-01-28

**Authors:** Handajaya Rusli, Rindia M. Putri, Anita Alni

**Affiliations:** 1Analytical Chemistry Research Division, Faculty of Mathematics and Natural Sciences, Institut Teknologi Bandung, Jl. Ganesha No. 10, Bandung 40132, Indonesia; 2Biochemistry Research Division, Faculty of Mathematics and Natural Sciences, Institut Teknologi Bandung, Jl. Ganesha No. 10, Bandung 40132, Indonesia; 3Organic Chemistry Research Division, Faculty of Mathematics and Natural Sciences, Institut Teknologi Bandung, Jl. Ganesha No. 10, Bandung 40132, Indonesia

**Keywords:** liquid chromatography, stationary phase, proteins, chiral molecules, PAHs

## Abstract

Compound separation plays a key role in producing and analyzing chemical compounds. Various methods are offered to obtain high-quality separation results. Liquid chromatography is one of the most common tools used in compound separation across length scales, from larger biomacromolecules to smaller organic compounds. Liquid chromatography also allows ease of modification, the ability to combine compatible mobile and stationary phases, the ability to conduct qualitative and quantitative analyses, and the ability to concentrate samples. Notably, the main feature of a liquid chromatography setup is the stationary phase. The stationary phase directly interacts with the samples via various basic mode of interactions based on affinity, size, and electrostatic interactions. Different interactions between compounds and the stationary phase will eventually result in compound separation. Recent years have witnessed the development of stationary phases to increase binding selectivity, tunability, and reusability. To demonstrate the use of liquid chromatography across length scales of target molecules, this review discusses the recent development of stationary phases for separating macromolecule proteins and small organic compounds, such as small chiral molecules and polycyclic aromatic hydrocarbons (PAHs).

## 1. Introduction

Separation is a critical process to isolate a particular compound, whether it is a natural product or a synthetic product. Studies of a compound’s characteristics and elucidation structure provides reliable results for pure compounds because there is no interference from other compounds. The primary source of difficulty in a separation process is the high similarity between two or more compounds, such as racemic and homologous mixtures. Liquid chromatography has proven to be an effective solution to those problems. The key to liquid chromatography separation is a sustainable retention and elution process. Stationary phases essential for separating compounds in liquid chromatography. Various liquid chromatography columns of both preparative and quantitative types have been used and continue to develop. For this reason, multiple studies and publications related to liquid chromatography can be found and accessed easily.

This review will discuss recent developments in liquid chromatography stationary phase technology for compound separation. Analytes that were once inseparable can now be separated and appropriately analyzed. The discussion focuses on the development of column material and separation interactions. This development aims to optimize and overcome the lack of existing columns. This review will discuss three types of analytes to demonstrate the use of liquid chromatography across length scales of target molecules, which are (1) larger biomolecules, particularly the proteins, (2) small chiral molecules, and (3) polycyclic aromatic hydrocarbons (PAHs).

## 2. The Principle of Separation of Compounds in Liquid Chromatography

Separation using liquid chromatography is possible because of the different interactions between the compounds present in the sample with the stationary and the mobile phases in the liquid chromatography system (Figure 1). Stationary phases can be developed to enable compound separation based on several modes, such as: (A) differences in the affinity to the compounds, (B) differences in the strength of electrostatic forces with the target compounds, and (C) size differences of target compounds. One or more of these modes of interaction will result in compound separation using liquid chromatography. Choosing the appropriate columns, whether commercial or under development, could be somewhat confusing for a user. Understanding the modes of interaction will be helpful to assist users in selecting the appropriate columns for a particular type of analyte.

In affinity-based separation (Figure 1A), stationary particles (black) with specific functional groups (red) interact with the appropriate compounds (green) through specific binding via a network of interactions, resulting in retention in the column. The stationary phase does not suspend noninteracting compounds (purple), and the compounds flow without retention. The common interactions in this mode of separation are hydrogen bonding, dipole–dipoles interaction, London force, and complex formation. The combination of hydrogen and dipole–dipole interaction facilitates a hydrophilic interaction, while the London force is a hydrophobic interaction.

Complex formation commonly occurs in chromatography where the analytes are proteins [1,2,3], DNA [4], and RNA [5,6]. The stationary phase is modified so that it has a metal-complex binding site that can accept ligands from macromolecules. A stronger interaction between compounds and the stationary phase increases retention time, while a weaker interaction decreases retention time. It is almost impossible to find compounds that have no affinity interactions at all. Even though stationary particles and compounds have very different polarities and no functional groups can interact, London forces always exist. Each molecule interacts differently due to their different functional groups, atomic numbers, and 3D structures. This difference allows for the separation of analogue [7] and homologous compounds [8,9].

Affinity chromatography is commonly employed in the separation of biomacromolecules. Further, reverse-phase chromatography (RPC) and hydrophilic interaction liquid chromatography (HILIC) are common for the separation of smaller organic compounds, in which affinity-based separation occurs as a result of interactions based on hydrophobicity and hydrophilicity.

The mode of separation that is based solely on electrostatic separation requires coulombic forces to occur between cationic species and anionic species. Stationary particles can function either as cations or anions, depending on the type of functional groups and the experimental conditions. As exemplified in Figure 1B, the blue-colored anion, which binds to the stationary cation (black) via electrostatic forces, is replaced by the analytes (red) due to a stronger interaction. This process is an ion-exchange mechanism. Green-colored compound, which is neither charged nor negatively-charged, is eluted from the column without being retained, as no coulombic interaction occurs between the green-colored compound and the stationary cation. The same principle applies for the separation of cationic compounds using anionic stationary phases. A greater charge density of molecules will produce a stronger electrostatic force, resulting in a longer retention time.

Such electrostatic forces are widely used in ion exchange chromatography (IEC) or ion chromatography (IC) systems. The charge of the stationary phase and analyte can be changed by adjusting the pH or composition of the mobile phase system so that separation can be carried out. The pH setting affects molecules that are weak acid or weak base. The new temporary molecules or the changes of 3D conformation can result from the changes of the mobile phase. The variables that can change in a mobile phase system are salinity, buffer type, and complexing ligand. The types of samples sample are small ions [10,11] and macromolecules [12].

Separation based on size difference is better known as size exclusion chromatography (SEC) or gel permeation chromatography (GPC). In this type of chromatography, compounds are separated based on the possibility of the analyte being trapped in stationary particle pores. Commonly, the smaller sized analyte is retained for a longer time in the stationary pores, so that the retention time is increased compared to bigger compounds, as shown in Figure 1C. Generally, this system is used to separate compounds with large molecular masses between 2000 and 20,000,000 amu, such as proteins [13,14], carbohydrates [15], surfactants [16], and polymers [17].

Combining two or three of the above modes of separation could result in more effective compound separation compared to applying only one mode of separation. For instance, chiral chromatography combines the principles of size exclusion and affinity-based separation. Electrochromatography (EC) [18,19,20], widely used to separate proteins and DNA, combines size and electrostatic separation principles. Separation in EC is based on the ratio of the charge density of the compounds. Larger charge density compounds elute faster than the low ones. Mixed-mode chromatography (MMC) [8,21,22,23,24,25,26,27,28,29,30,31,32,33,34,35,36,37] is a system that can be used for at least two types of chromatography depending on the measurement conditions, especially in the mobile phase. Multi-column chromatography (MCC) is used to separate compounds from complex matrices utilizing a combination of the three separation principles above.

Table 1 summarizes several examples of the types of chromatography that have been developed in the last 10 years. Most of the developed stationary phases are based on silica and organic polymers. The samples analyzed are also extensive, ranging from simple ions to uncharged macromolecules, from polar to nonpolar molecules, and from water-based to organic-based solvents. Various isomers (functional group, optical, and structural isomers) or molecule derivatives can be separated using an appropriate liquid chromatographic system.

The table also shows that most mobile phase types are water, acetonitrile, and methanol. Several modifications by adding organic or inorganic acids and bases were carried out to condition the pH and affinity in the chromatographic system. This selection depends on to type of liquid chromatography will be applied. Theoretically, the combination of mobile phases that can be used is unlimited. The compatibility between stationary and mobile phase needs to be carefully considered. The mobile phase mixture must be completely miscible with each other. The mobile phase must not damage or dissolve the stationary phase.

## 3. Development of Stationary Phases for Selective Protein Capture and Purification

Like the separation of chiral compounds, selective binding is also an essential component in the separation of larger biomolecules, and particularly of the proteins. Often, selective binding is achieved with the use of affinity chromatography, which is characterized by stationary phases that capture the protein selectively via a network of carefully crafted intermolecular interactions. In order to establish such a selective binding of proteins, a particular affinity ligand is attached to the surface of the stationary phase. Then, surface-immobilized ligands act as binding sites and capture target proteins via specific interactions, including electrostatic interaction, coordination complexes, and protein–ligand binding. Various affinity columns for proteins have been developed to increase selectivity, tunability, and reusability.

In recent years, various stationary phases have been developed to target specific proteins. One of the main approaches for selective protein binding is using metal ions to capture a tagged protein. For instance, a poly-His-tagged protein provides chelating ligands through the amine moieties in histidine residues. Such systems have been widely used for affinity chromatography in protein purification, in which nickel(II) ions are immobilized at the surface. As well as nickel ions, cobalt ions have recently been developed to capture various proteins, such as fused xylanase interacting with a scaffolding protein [52].

Wong et al. designed a hemicellulose-based stationary phase decorated with Co^2+^ ions (Figure 2) to bind a scaffold protein, CipA, bearing a cohesin unit that could selectively capture dockerin-fused xylanase (XynCt). The hydroxyl groups from hemicellulose were modified with epichlorohydrin (EPI) and iminodiacetatic acid (IDA) to produce a dicarboxylic acid amine. These groups allowed complexation with Co^2+^ ions. The immobilized ions further bound His-tagged CipA via covalent coordination. Then, the cohesin unit of CipA selectively docked the dockerin unit fused into the target enzyme (i.e., xylanase). A similar cobalt-based system was also employed to capture bovine serum albumin using modified chitosan [53]. Furthermore, a combination of ions can also enhance selectivity to His-tagged proteins, for instance, copper(II) ions and nickel(II) ions that are co-immobilized on a crosslinked chitosan/polyvinyl alcohol support [54].

The immobilized ions are usually stabilized by chelating ligands, such nitrilotriacetic acid (NTA), which are covalently anchored to the stationary phases. Several strategies have been reported to efficiently incorporate NTA ligands into the membranes through click-chemistry to azido-containing cellulose acetate membranes [54]. Another example of chelating ligands is hydroxamic acid. This ligand can stabilize copper ions immobilized onto polymethylmethacrylate-grafted cellophane membranes. Notably, the hydroxamic acid-Cu^2+^ system showed no leakage of Cu^2+^ ions upon elution with 88% protein His-tagged chitinase (2).

As well as nickel and copper, ions like zirconium(IV) and iron(III) have emerged as alternatives for improving binding selectivity. He et al. reported a metal affinity stationary phase with phosphate-Zr^4+^ moieties. This stationary phase selectively captures phosphoproteins (casein and ovalbumin) in nonphosphoprotein mixtures (albumin and lysozyme) [55]. Lysozyme was proven to effectively bind with an iron(III)-iminodiacetic acid chelating ligand system with up to 365 mg of lysozyme per 1 g of regenerated cellulose nanofibrous membranes and a 4-h adsorption time [56]. The proteins that bind via metal complexation could be released from the stationary phase by adding competitor ligands such as imidazole to the mobile phase. This method is commonly used in the purification of His-tagged proteins. Riguerro et al. identified the metal binding capacity and ratio of polyhistidine tag residues to determine the chromatography performance. Imidazole 250 mM was gradually added into the mobile phase to completely elute 4 and 6-His-tagged proteins out of the affinity columns [3].

Organic molecules may also act as a selective affinity ligand for protein capture. A specific ligand for lysozyme such as diethyl-4-aminobenzyl phosphonate can be grafted to macroporous membrane supports [57]. Tris(hydroxymethyl)-aminomethane can increase lysozyme purity by over 100-fold when anchored onto a polyacrylonitrile nanofiber membrane [58]. Furthermore, lysozyme is also capable of binding dyes. Therefore, dye molecules can be immobilized onto the surface to bind lysozyme selectively. It was exemplified in a study using host–guest chemistry, in which cyclodextrin molecules were introduced into a membrane made of clickable ethylene-vinyl alcohol (EVAL). Figure 3 shows that the azide-bearing cyclodextrin could attach to the surface of EVAL through click-chemistry, and subsequently bound dye molecules via host-guest chemistry. The dyes acted as ligands for the target biomolecule, which in this case was the lysozyme [59].

One of the main challenges in developing selective stationary phases to capture proteins is ensuring that the proteins do not lose their structural integrity and activity upon binding and elution. Affinity stationary phases based on metal ion immobilization have been reported to be reusable and do not significantly alter the bound enzyme activity, even after multiple cycles of binding and elution [52,60]. Similarly, immobilization of specific ligands on the surface has also been presented as a feasible strategy without compromising the enzyme activity. For instance, a study showed that attachment of Cibacron Blue F3G-A dye on cellulose membrane to selectively immobilize amylase led to an enhancement of starch hydrolysis on the surface [61].

Another challenge in this area of research is selectively binding one protein in the myriads of other proteins, such as in blood plasma. Tuning the surface properties and choosing affinity ligands that do not allow nonspecific interactions are practical approaches to addressing this issue. For instance, the high selectivity of human IgG over human serum albumin was achieved using alginate dialdehyde layers on a nylon membrane, which significantly reduced nonspecific adsorption [62]. Moreover, peptides that selectively bind monoclonal antibodies, such as the immunofibers, could also be introduced on the stationary phase surface [63].

Emerging targets for selective capture and purification using affinity chromatography are blood proteins such as albumin and antibodies (immunoglobulins). It has been shown that albumin can be selectively adsorbed onto poly(2-hydroxyethyl methacrylate) via cholic acid as the affinity ligand [64]. Aside from small organic molecules, polypeptides also have potential as affinity ligands for protein capture. Sericin, a silkworm polypeptide, upon blending with chitosan membranes with a ratio of 4:1, was capable of recognizing serum albumin and led to a 45% increase in adsorption capacity compared to chitosan alone [65]. For antibody purification, an alternative for affinity ligands is tryptamine, which can bind the nucleotide-binding site on the Fab domain of the antibodies [66]. This strategy created both monoclonal and polyclonal antibodies purified from cell culture media of ascites fluids.

In a different study, selective binding of immunoglobulin G (IgG) was shown to be affected by the length of the chemical spacer immobilized onto the cellulosic membrane [67]. Remarkably, even fully folded, complex protein chains, such as Protein G on a cellulose surface, can be utilized as affinity ligands which selectively binds the interleukin-6 (IL-6) antigen [68]. Another common protein of interest among blood proteins is the plasminogen used in eye treatments including eye surgery. L-lysine, an affinity ligand, was functionalized to regenerated cellulose membranes effectively bind the plasminogen [69].

Apart from affinity chromatography, hydrophobic charge-induction chromatography (HCIC) can be used to purify antibodies selectively [46]. HCIC is characterized by a combination of hydrophobic and electrostatic interactions between the stationary phase and the target protein, which are adjustable with pH change. Lu et al. reported a resin made of 2-mercapto-1-methyl-imidazole (MMI)-modified agarose beads that preferentially captured IgG over albumin, yielding a 95% pure IgG. The optimized adsorption of IgG was achieved under loading at pH 7.0 and elution at pH 4.0 using MMI resin with a pore size of 43.2 nm and density of 101 μmol/g gel. A comparison between affinity chromatography and HCIC was conducted in the context of purification of a monoclonal antibody against Ebola GP1 protein [70]. Fulton et al. demonstrated that Protein A is a ligand that displays a high affinity toward antibody binding, resulting in an 85% recovery of the target protein. Meanwhile, the combination of hydrophobic interaction chromatography (HIC) and HCIC resulted in an average recovery of only 36% under elution at pH 5.0.

Mixed-mode chromatography (MMC) has also been developed for better protein purification. In an MMC setup, as shown in Figure 4, two or more binding modes are employed and combined to improve selectivity and protein recovery. Xiong et al. reported using 2-methacryloyloxyethyl phosphorylcholine (MPC) as a ligand anchored to silica through a thiol–ene click reaction to prepare a stationary phase for MMC [35]. Due to zwitter-ionic MPC, a combination of hydrophilic interaction chromatography (HILIC) and weak cation-exchange chromatography (WCX), take place upon protein binding and elution separating proteins based on hydrophobicity and charges simultaneously.

Phosphorylcholine (PC)-based ligands have been reported to interact with proteins via multiple modes, such as electrostatic adsorption, electrostatic repulsion, and hydrophobic interaction [35]. Compared to PC ligands, MPC ligands were reported to add hydrophobic effect and hydrogen bonds, which may contribute to the enhanced protein captures and improved stability of their stationary phase. A recombinant human Delta-like1-RGD (rhDll1-RGD) expressed in *Escherichia coli* was collected to a 63.4% recovery and 97% purity, respectively, via one-step purification using this system. Remarkably, the developed MMC could handle simultaneous purification and refolding of the recombinant protein.

As recombinant protein technology has become more advanced, in conjunction with other techniques, chromatography has been developed into a simultaneous purification and refolding system for insoluble proteins, particularly the inclusion bodies frequently obtained during recombinant protein overexpression. Ye et al. reported the recovery of a fibroblast growth factor-21 (FGF-21) protein recombinantly expressed in *E. coli* Rosetta (DE3) as inclusion bodies. Large-scale purification and refolding of FGF-21 was achieved using double Ni-affinity chromatography with a hollow fiber membrane (HFM) column [71].

Aside from affinity columns, mixed-mode chromatography (MMC) using stationary phases containing sulfonic acid ligands has been proven effective in recovering refolded proteins expressed in the form of inclusion bodies, such as human bone morphogenetic protein-2 (rhBMP-2) expressed in *E. coli*. In a study by Gieseler et al., a one-step MMC was shown to selectively purify the rhBMP-2 dimers from the refolding mixtures, yielding high biological activity comparable to commercial products. Notably, MMC remained effective in an environment with high ionic strength, such as the refolding buffers [72].

## 4. Development of Stationary Phases for Small Chiral Compounds

Among small organic compounds, chiral molecules are particularly interesting targets for the development of stationary phases. This is due to the fact that enantioselectivity is an essential and characteristic feature of drug–receptor interactions in the human body, and impure enantiomeric drugs have been reported to result in a high fatality rate. An example of this is the thalidomide tragedy, which prompted scientists to address chiral molecule separations very seriously [73]. Although chemical modifications can separate the mixture of enantiomers, direct chromatographic resolution of enantiomers is preferred to avoid unnecessary derivatization, significantly reduce sample manipulation, and increase the speed of obtaining results obtained after separation by chromatography. LC using chiral stationary phases (CSPs) is the most widely used technique for the enantioselective separation of chiral drugs [74].

The choice of stationary phases in chromatographic columns is a crucial step in molecule separation. Silica has long been regarded as a material of choice in the separation of small molecules due to a comprehensive understanding of its surface chemistry and surface modification. However, its relative thermal and chemical instability limit its use in the development of efficient separations for complex molecules. Si-O-Si bond hydrolyzes at pH > 8, at elevated temperatures (>40 °C), and in the presence of very common but highly deleterious species such as phosphate and carbonate. It is soluble in aqueous/organic media that is even slightly alkaline at even moderately elevated temperatures. The siloxane bond is unstable at an acidic pH and becomes increasingly less stable as the pH is lowered [75].

Polymeric reversed phases are stable from pH 1 to 13 despite displaying excessive swelling and low mechanical stability under common use conditions. Developments of polymeric materials such as polystyrene based materials with various modifications have been rapid [76]. Svec and Maya (2014) prepared poly(styrene-*co*-divinylbenzene) monoliths with various crosslinkers and explored their chromatographic performance in separating small chiral molecules. Modifications of the polymer were carried out by Friedel–Crafts reaction using Fe^3+^ as a catalyst. They observed that the reaction conditions, namely (i) the temperature and time used for polymerization of the precursor monoliths, (ii) the amount and loading procedure of the external crosslinker and catalyst, and (iii) the temperature and reaction time of the Friedel–Crafts alkylation affected the monoliths’ properties and hence their chromatographic performance. Applications of polymeric materials based on polystyrene have also been reported in the separation of alkyl aryl ketones and barbiturates under reversed-phase high-performance liquid chromatography (RP-HPLC) at a high temperature (100–200 °C) [77].

In recent years, polysaccharide and protein-based materials have been developed and received considerable attention to introduce chirality to the stationary phase materials. As a naturally chiral and abundantly available material, polysaccharide and protein modification are easy to use, which affirms their potential as liquid chromatography stationary phase materials. Cellulose is one class of polysaccharides that can be developed as a chiral stationary phase due to its strong chiral recognition ability, large loading capacity, and ease of functionalization, which make modification for specific targets possible.

Cellobiohydrolase (CBH) is one type of protein that can act as a chiral selector, which, upon immobilization onto silica gel, was developed as an aromatizing chiral column and is available commercially as part of Chiralpak CBH [78]. Fu et al. evaluated two kinds of columns, Lux Cellulose-1 and Chiralpak CBH, to separate mixtures of enantiomeric drugs, namely β-blockers, antacid, and cathinone. Between the two, the Chiralpak CBH column baseline separated six chiral drugs using reversed systems, which had mobile phase composed of 5 mM ammonium acetate aqueous (pH = 6.4)/methanol (95/5, *v*/*v*).

For identification purposes, a common method for analytical separation is arranging a tandem instrument, such as reported by Zhao et al. in the separation of pesticide mixture from a soil sample [79]. Chiral liquid chromatography–tandem mass spectrometry was able to analyze the enantiomeric composition of the eight chiral pesticides in solid environmental matrices after a series of sample preparations, as shown in Figure 5. It is notable that the separation of S-metalaxyl and R-metalaxyl was achieved using polysaccharide-based column (Chiralcel) aqueous formic acid (0.1%, *v*/*v*) (A) and acetonitrile (B) performed in isocratic mode (60:40, *v*/*v*) at a flow rate of 0.6 mL min^−1^.

Another strategy in developing the stationary phase for chiral compound separations is using extensively reviewed Pirkle-type chiral stationary phases (CSPs) based on chiral recognition by a chiral selector normally composed of small molecules. The choice of naturally abundant small molecules based on CSPs is preferred. Derivatives of phenylglycine and leucine, in addition to quinine, have been extensively explored as CSPs [80,81].

Cyclodextrin is a common chiral selector for chiral LC column preparation. Recently, Malsche et al. reported the application of this compound as coating in porous silicon pillar array column and exploring its chromatographic performance in separation of chiral nonsteroidal anti-inflammatory drugs (NSAIDs) [82]. Interactions of enantiomers with the hydroxypropyl-β-cyclodextrin (Hp-β-CD) stationary phase resulted in different retention times for each enantiomer, which further enabled its efficient separation. There are two methods for preparing CSPs via coating technique, namely chemical and dynamic coating, with each having benefits and disadvantages. The former is superior in the enantioseparation of amino acids and dipeptides, while the latter resolves α-hydroxy acids better [83].

Recent chiral LC stationary phase development trends have included applying chiral metal–organic frameworks due to their adjustable cavities and abundant chiral active sites. However, some challenges such as wide particle size distribution and irregular shape of MOFs, which result in low column efficiency, undesired chromatographic peak shape, and high column backpressure are yet to be overcome and remain research topics of interest. One attempt at overcoming these challenges was reported by Zhang et al. who fabricated chiral core-shell microspheres (Cu_2_(d-Cam)_2_(4,4′-bpy))_n_@SiO_2_ composite as CSP and used it to separate racemates including alcohols, amines, ketones, epoxides, and organic bases successfully [84]. The column exhibited superior stability and repeatability for the separation of chiral compounds.

The one-pot synthesis method for the immobilization of chiral MOF (Cu_2_((+)-Cam)_2_Dabco) (Cu_2_C_2_D) onto microspherical silica particles further simplified the preparation of the MOF-based chiral stationary phase [85]. This method offers a uniform core-shell microsphere, tunable shell thickness, and corresponding column efficiency. The chiral LC stationary phase prepared by this method was applied to separate the product of asymmetric Michael addition reaction with baseline separation of both enantiomers Figure 6. Figure 6 shows the fabrication scheme of SiO_2_@Cu_2_C_2_D by modification of silica (SiO_2_) particles with copper (Cu(AcO)_2_·2H_2_O), D-(+)-Camphoric acid (D-Cam), and 1,4-diazabicyclo[2.2.2]octane (Dabco) to introduce the Cu_2_C_2_D shell with a density that can be controlled by repeating treatment procedures to obtain the best enantiomer separations.

Ionic liquids (ILs) are promising new materials in chiral small compound separations. ILs have Janus behavior, which can separate nonpolar compounds as if they were nonpolar stationary phases and separate polar compounds as polar stationary phases. In the application as CSPs, chiral ILs can be used directly, or achiral ILs can be used as a solvent for other chiral separators. Separation by chiral ILs is achieved via ion exchange interaction between the liquid cationic groups and anion samples, such as in the separation of chiral acids. He et al. prepared a chiral imidazolium by ring-opening cyclohexene oxide with imidazole derivatives followed by modifying substituents.

Furthermore, the chiral ILs were immobilized onto the surface of the silica sphere through click chemistry [86]. It was observed that the substituent significantly influenced chiral separation abilities on the chiral selector as well as the mobile phase, which indicated that other ion exchange interactions existed. By exploring various parameters in the experiments, it was suggested that steric hindrance, hydrogen bonding, and π-π interaction are essential for enantioselectivity.

Chiral imidazolium ILs bonded to silica as CSPs, and their application for the separation of organic acids was also reported by Huang et al. [87]. The column material was immobilized on silica and showed their potential application in the separation of chiral pharmaceuticals. Figure 7 exhibits the schematic representation of chiral ILs as column materials impregnated onto silica. Variations of substituents (R) enable tuning of chromatographic performance towards certain racemic drugs. A thorough exploration of the LC conditions suggested that high acetonitrile content in mobile phases was conducive to enantiorecognition for this stationary phase.

## 5. Development of Stationary Phases for PAHs Capture

Another class of small organic compound that is an emerging target to be efficiently captured using LC stationary phases is polyaromatic hydrocarbons (PAHs), as they are leading pollutants causing environmental issues, second only to microplastics. In general, PAHs have a characteristic structure that is rich in aromatic groups. Some examples of PAHs that are often studied are naphthalene, anthracene, fluorene, pyrene and its derivatives, and fluoranthene and its derivatives. The structures of the PAHs are shown in Figure 8.

PAHs can be analyzed using various methods, namely HPLC [88,89], GC [90,91], and voltammetry [92]. The types of samples analyzed have not been limited to water samples [93], and have also included dietary products [94] and beauty products [95]. The PAHs in the samples generally have very low concentrations and very complex matrices. Therefore, the PAH analysis of real samples always begins with the extraction process, which is either solid-phase extraction (SPE) or liquid–liquid extraction (LLE).

Analysis using HPLC is the most common technique for PAHs because it can analyze all types of PAHs. GC cannot analyze PAHs with high boiling points such as indeno(1,2,3-cd)pyrene with a boiling point of 536 °C, while voltammetry cannot distinguish signals from PAHs with similar structures. The most standard HPLC column used to analyze PAHs is silica with an octadecyl functional group on its surface, better known as column C18 [96,97,98]. The standard stationary phase does not provide satisfactory results for a sample composed of many PAHs. Adding alkyl-imidazolium ionic liquid did not provide the expected result [99]. Therefore, the stationary phase needs to develop using new materials.

The most commonly developed and commercially available stationary phase for PAH analysis is silica grafted with carbon chains with certain functional groups. To prepare the silica particles, the precursors commonly used are tetramethoxysilane (TMOS), tetraethoxysilane (TEOS), and sodium silicate. As PAHs display hydrophobic characteristics, they require a hydrophobic stationary phase to establish an interaction. Silica has hydrophilic properties due to the presence of silanol groups, so it is theoretically unsuitable for PAH separation. Therefore, the silica surface must be made hydrophobic using a modifier.

Some of the modifiers that have been used are trichloro-octadecylsilane (TCES), 4-aminobutyldimethylmethoxysilane (ABDMS), vinyltrimethoxysilane (VTMS), and phenylpropyldimethylchlorosilane (PPDMC) [100,101,102,103]. The modifiers that have been developed have similarities, namely the presence of alkoxyalkylsilane and halide alkylsilane families. The modifier reacts with most of the silanol groups to obtain hydrophobic silica. To increase the hydrophobicity of silica, it is necessary to carry out an end-capping process. This process aims to remove remaining silanol groups. A common feature of end-capping agents is a tertiary N-base such as triethylamine, hexamethyldisilazane, and *N*-(trimethylsilyl)dimethylamine. Smaller coupling agents, such as trimethylchlorosilane and trimethylethoxysilane, can also be employed as end-capping agents. 

Taskin et al. used TCES to obtain C18 silica particles. Unlike previous research, this study used pyridine addition, which is a byproduct of the reaction between TCES and silica particles, as an HCl binder. End-capping was performed using hexamethyldisilazane [100]. A slightly more complex method was used by Mermat et al. This group made a series of reactions to attach a citronellol to a silica surface. Citronellol was bonded to the silica surface using the coupling agent ABDMS and 4’-hydroxy-4-biphenylcarboxyl. End-capping was performed using *N*-(trimethylsilyl)dimethylamine [101].

Stevenson et al. used phenylpropyldimethylchlorosilane and trimethylchlorosilane as coupling and end-capping agents, respectively. The study showed that increasing the amount of bound phenyl was able to increase the efficiency of separating PAHs [103]. Carbon dots (CD) are also able to provide better separation efficiency than commercial columns for the equivalent C18 type, as shown by Wu et al. CD was prepared from citric acid and octadecanamine using a solvothermal technique. The appropriate polarity CD was then attached to the silica surface using an EDC/NHS coupling agent [31]. However, in this study, no end-capping process was carried out, which would increase the separation efficiency.

In addition to silica-based systems, zirconia oxide can also be used as a support for the liquid chromatography stationary phase. Yan et al. used UiO-66 (a type of zirconium MOF) to analysis PAHs. The MOF was able to separate some PAHs but was not efficient [104]. Zhao et al. then synthesized UiO-67 by replacing the terephthalic acid precursor in UiO-66 with 4,4’-biphenyl-dicarboxylic acid. The results obtained, like those for other silica-based columns, were not as high as expected [41]. In both systems, the normal phase liquid chromatography system was used.

Particle-based acrylics have become one of the most common methods of PAH separation in recent years. Zhao et al. were able to produce poly(*N*-isopropyl acrylamide) on the surface of VTMS-modified silica using the principle of radical polymerization [102]. Increasing the number of reacted *N*-isopropyl acrylamide monomers increases the hydrophobicity of the particles. The stationary phase is suitable for application in a reverse-phase liquid chromatography system. Modification with acrylamide can also be carried out using 3-trimethoxysilylpropyl methacrylate on the silica surface. Demir et al. utilized a thiol–ene addition reaction to attach long chain alkyls to 3-trimethoxysilylpropyl methacrylate and increase the hydrophobicity of silica particles [105].

Like thiol–ene reactions, Michael addition reaction of thiol-methacrylate could also be used in the manufacture of silica columns. Lin et al. prepared a capillary column for liquid chromatography by reacting methacrylate-polyhedral oligomeric silsesquioxane with three multi-thiol crosslinkers at once, namely 1,6-hexane dithiol, trimethylolpropane tris(3-mercapto propionate), and pentaerythritol tetrakis (3-mercapto propionate) with dimethyl-phenyl phosphine as a catalyst [106]. Mallik et al. reacted octadecyl acrylate, *N*-methyl maleimide, and 3-mercaptopropyl trimethoxysilane using an AIBN catalyst to obtain a telomere. This telomere was then hydrolyzed on the silica surface [39].

The developed acrylate-based stationary phases are not limited to silica matrixes; but they also extend to organic polymers. The advantage of using nonsilica particles is that there is no interference from the silanol group, which tends to be hydrophilic. Rosin developed by Wang et al. showed a good ability to separate PAHs. Rosin was synthesized by a monomer methyl methacrylate and ethylene glycol dimethacrylate, a crosslinker acryl pimaric acid ethylene glycol acrylate, and an AIBN catalyst [107]. Peng et al. also succeeded in synthesizing a capillary column from the polymerization reaction of 12-methacryloyl dodecylphosphatidic acid and ethylene glycol dimethacrylate with an AIBN catalyst [45]. Yu et al. reported similar results but used propargyl methacrylate and ethylene dimethacrylate as catalysts [108].

## 6. Outlook

As liquid chromatography continues to be one of the main tools in the preparation and analysis of chemical compounds, various developments have been made to improve the binding selectivity and reusability of the columns. The surface chemistry of the stationary phases of silica, polysaccharides, and synthetic polymer matrixes has been adjusted and modified to result in better separation across length scales. The stationary phases can be chemically decorated with desired functional groups to ensure a stable and selective binding to the target molecule. For selective separation of chiral molecules, chiral stationary phases (CSPs) have been prepared using modified polystyrene, cellohydrobiolase, cyclodextrins, chiral metal–organic frameworks (MOF), and ionic liquids as chiral selectors. Hydrophobic surfaces are an essential component of liquid chromatography for PAH binding. Several hydrophobic chemical modifiers have been covalently anchored at silica surfaces to increase their affinities towards PAHs. Moreover, hydrophobic MOF and particle-based acrylics have also been developed as stationary phases to capture PAHs efficiently. In the case of protein capture and purification, affinity chromatography is commonly established through metal–protein complexation as well as ligand–protein complexation. Furthermore, by adjusting the conditions of elution, particularly the pH and ionic strength, hydrophobic charge-induction chromatography (HCIC) and mixed-mode chromatography (MMC) have also been investigated to achieve better protein purification.

Despite the recent progress, there is still room for improvement in the advancement of liquid chromatography stationary phases. Novel materials can be developed into potential adsorbents as stationary phases. The popularization of click chemistry and host–guest chemistry presents opportunities for the modification and diversification of stationary phase materials. Chromatography has been proven effective in isolating specific target molecules at a reasonable cost. Unfortunately, it is difficult to achieve this due to a lack of suitable equipment and systems on an industrial scale. Thus, the challenge of expanding such systems into large-scale setups, such as industrial preparations, in which reusability and scalability are required remains.

## Figures and Tables

**Figure 1 molecules-27-00907-f001:**
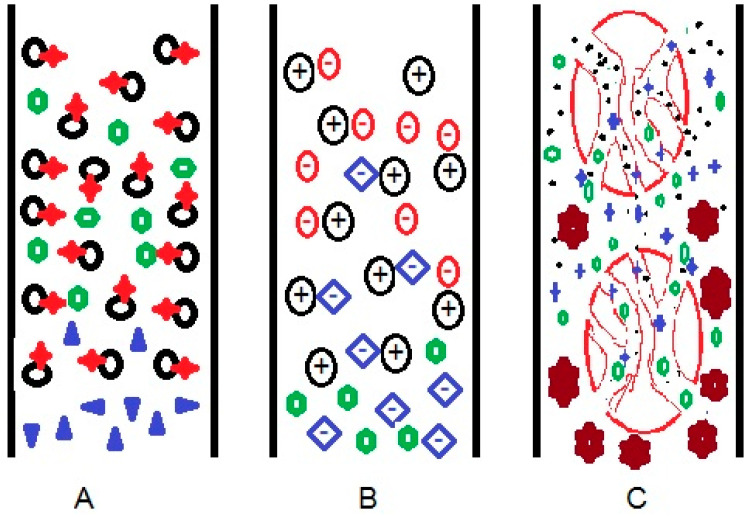
Scheme of physical interactions between column and target molecules, based on (**A**) affinity, (**B**) electrostatic forces, and (**C**) size difference in liquid chromatography systems.

**Figure 2 molecules-27-00907-f002:**
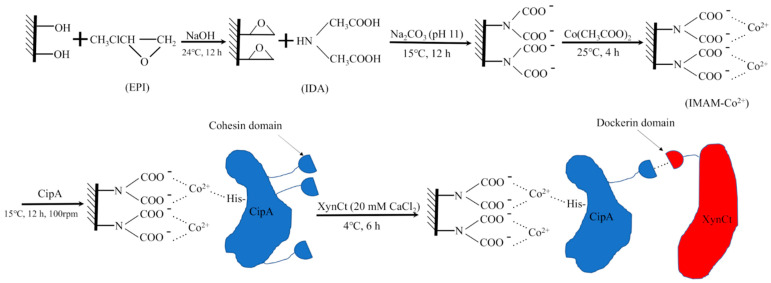
Modifications of hemicellulose using epichlorohydrin (EPI) and iminodiacetatic acid (IDA) into a Co^2+^-decorated stationary phase to bind His-tagged scaffold protein, CipA, bearing a cohesin unit that selectively captures a dockerin unit fused into a target enzyme (i.e., a xylanase). Reprinted with permission from ref. [52]. Copyright 2020 MDPI.

**Figure 3 molecules-27-00907-f003:**
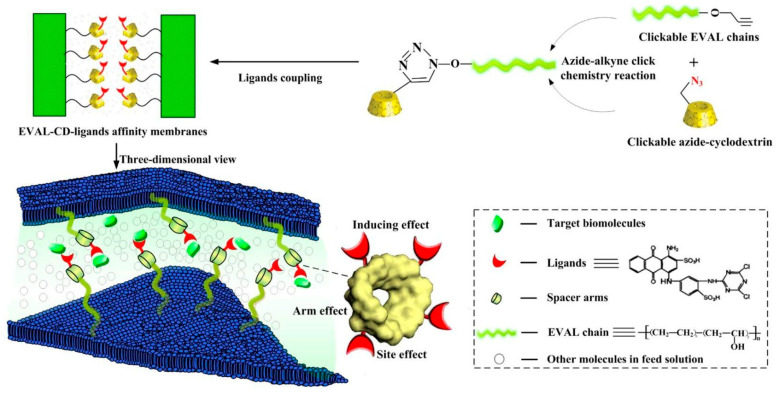
Click modification of ethylene-vinyl alcohol (EVAL) using surface-accessible cyclodextrin spacers, allowing capture of dye molecules, which subsequently act as a ligand for the target biomolecule, i.e., the lysosome, through host–guest chemistry. Reprinted with permission from ref. [59]. Copyright 2017 Elsevier.

**Figure 4 molecules-27-00907-f004:**
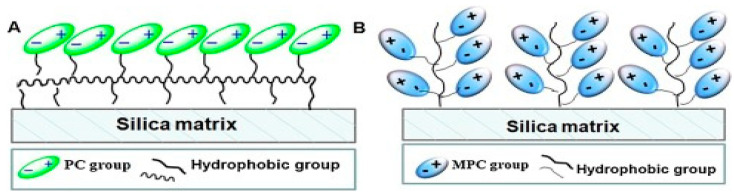
Illustration of mixed-mode chromatography (MMC) stationary phases, employing zwitter-ionic (**A**) phosphorylcholine and (**B**) 2-methacryloyloxyethyl phosphorylcholine as ligands anchored to silica matrix. Reprinted with permission from ref. [35]. Copyright 2018 Elsevier.

**Figure 5 molecules-27-00907-f005:**
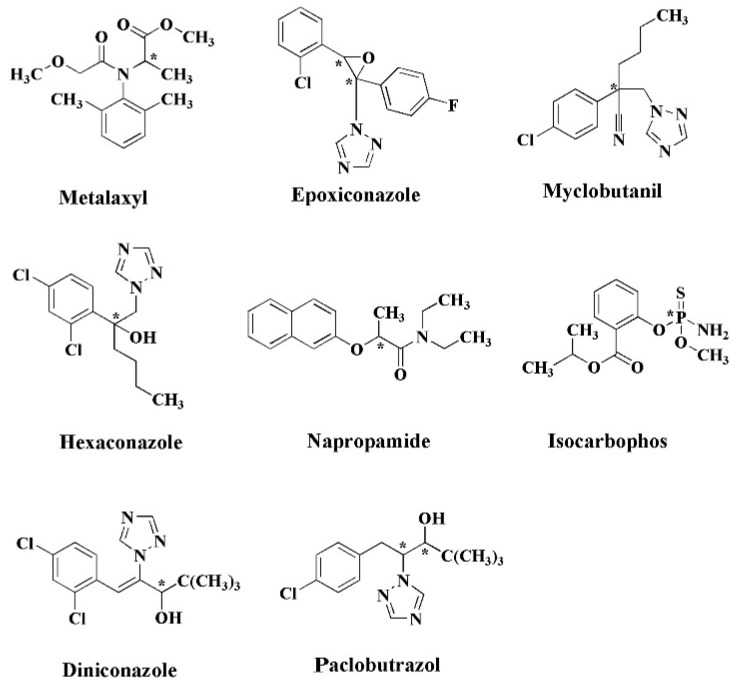
Chemical structures of chiral pesticides (* chiral center). Reprinted with permission from ref. [79]. Copyright 2018 Elsevier.

**Figure 6 molecules-27-00907-f006:**
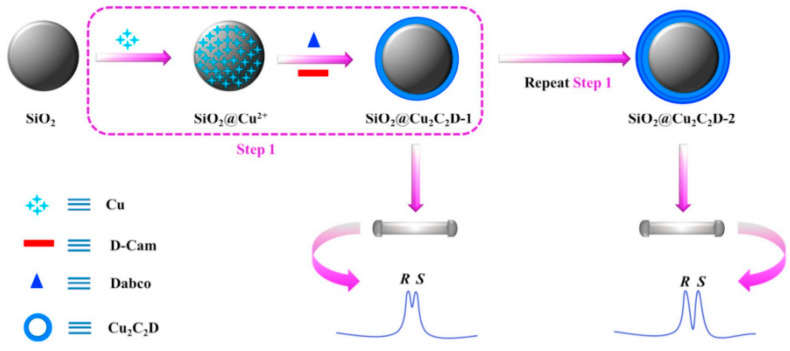
Fabrication of SiO_2_@Cu_2_C_2_D by modification of silica (SiO_2_) particles with copper (Cu(AcO)_2_·2H_2_O), D-(+)-Camphoric acid (D-Cam), and 1,4-diazabicyclo[2.2.2]octane (Dabco). Reprinted with permission from ref. [85]. Copyright 2018 Elsevier.

**Figure 7 molecules-27-00907-f007:**
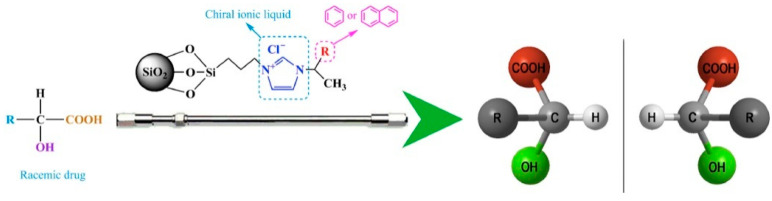
Chiral imidazolium stationary phases for HPLC. Reprinted with permission from ref. [87]. Copyright 2016 Elsevier.

**Figure 8 molecules-27-00907-f008:**
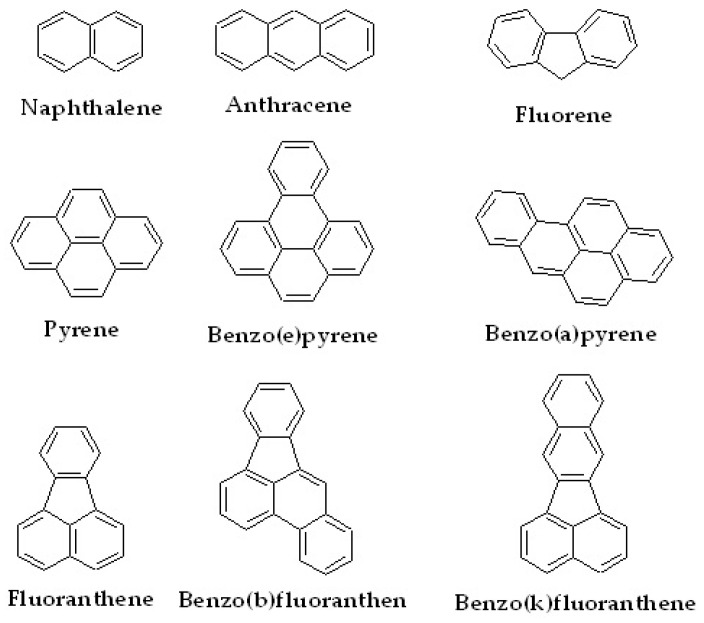
Several types of PAHs and their derivatives.

**Table 1 molecules-27-00907-t001:** Types of liquid chromatography, separation principles, and new developments in the field of liquid chromatography.

Mode of Liquid Chromatography	Separation Principle	Stationary Phase	Analyte	Mobile Phase
Reverse-phase chromatography	Affinity	Silica modified with octadecyl acrylate and 2-vinyl-4,6-diamino-1,3,5-triazine [38]	PAHs	Methanol
	Affinity	Silica modified with octadecyl acrylate and *N*-methylmaleimide [39]	PAHs and tocopherols	Mixed of methanol and water
	Affinity	Silica modified with *N*-Boc-phenylalanine and cyclohexylamine [40]	Phytohormones	Mixed of phosphate buffer and acetonitrile
	Affinity	Zr_6_O_4_(OH)_4_ MOF modified with 2-amino-terephthalic acid or 4,4′-biphenyl-dicarboxylic acid [41]	PAHs and aromatics compound	Mixed of methanol and water
Hydrophilic interaction liquid chromatography	Ionic	Silica modified with (2-(methacryloyloxy)-ethyl)dimethyl-(3-sulfopropyl)ammonium hydroxide or 2-methacryloyloxyethyl phosphorylcholine [42]	Mixed of toluene, formamide, dimethylformamide, and thiourea	Mixed of water and acetonitrile
	Affinity	Amino silica modified with polyhedral oligomeric silsesquioxane and acrylamide derivatives [34]	Nucleosides, organic acids, and β-agonists	Mixed of acetonitrile and ammonium formate solution
	Affinity	Silica modified with EGDMA and maltose [32]	Nucleobases and nucleotides	Mixed of water and acetonitrile
	Affinity	Silica modified with vinyl silsesquioxane and dithiothreitol [43]		
	Ionic and affinity	Silica modified with pyrazinedicarboxylic anhydrate [44]	Oligosaccharides, alkaloid, and organic acid groups	Mixed of acetonitrile and ammonium formate solution
Mixed-mode chromatography	Ionic and affinity	Silica modified with 2-methacryloyloxyethyl phosphorylcholine [35]	Protein and lysozyme	Mixed of acetonitrile, ammonium formate solution, KH_2_PO_4_ solution, NaCl solution
	Ionic and affinity	Silica modified with octadecyl and diol groups [8]	Aristolochic acid and derivatives	Mixed of formic acid and acetonitrile
	Ionic and affinity	Silica modified with glutathione [26]	Protein	Mixed of water, formic acid, acetonitrile
	Ionic and affinity	poly(12-methacryloyl dodecylphosphatidic acid-co- ethylene glycol dimethacrylate) [45]	Ketone aromatic, phenol and derivatives, small organic compounds	Mixed of ammonium formate solution and acetonitrile
	Affinity	Amino silica modified with octadecyl and carbon dots [31]	PAHs, nucleosides, and nucleobases	Mixed of water and methanol, acetonitrile, and ammonium acetate solution
Affinity chromatography	Ionic and affinity	Agarose modified with 2-Mercapto-1-methylimidazole [46]	Protein	NaOH solution
	Ionic and affinity	Sepharose modified with ligand complex [3]	Protein with histidine	Mixed of Tris buffer, sodium chloride, and imidazole
	Ionic and affinity	Silica modified with N-methylimidazolium ionic liquid [28]	Protein	Mixed of acetonitrile, trifluoroacetic acid, NaClO_4_ solution, KH_2_PO_4_ solution, and NaCl solution
	Ionic and affinity	Amino silica modified with glutaraldehyde [47]	Protein	Phosphate buffer
Ionic chromatography	Ionic	Bentonite modified with chitosan and cetyltrimethylammonium bromide (CTAB) [11]	Cr(III) and Cr(VI) solution	Nitric acid solution for Cr(III) and ammonia solution for Cr(VI)
	Ionic	Polystyrene-methacrylate derivatives modified with poly(amidoamine) [10]	Small anions like nitrate, sulfates, bromide, etc.	NaOH solution
Chiral chromatography	Size and affinity	Polysaccharide modified with 3-chloro-4-methylphenylcarbamate [48]	Paroxetine hydrochloride groups	Mixed of supercritical CO_2_, methanol, and ammonium acetate solution
	Affinity	Isopropylcarbamate cyclofructan 6 groups [49]	Methionine groups	Mixed of methanol, acetonitrile, acetic acid, and triethylamine
	Size	Silica modified with 3,3′- phenyl-1,1′-binaphthyl-18-crown-6-ether [50]	Amino acids and peptides	Mixed of perchloric acid solution, acetonitrile, and methanol
	Affinity	Poly(styrene-divinylbenzene) coated with chitosan [9]	Benzoin	Mixed of water and acetonitrile
Electrochromatography	Size and ionic	Poly(POSS-co-META-co-DMMSA) [18]	Benzoic acid, nucleosides, bases, glycopeptides	Mixed of phosphate buffer, triethylamine, and acetonitrile
	Affinity	Silica modified with a metal-organic framework (MOF) [20]	Benzenes and derivatives	Mixed of phosphate buffer and acetonitrile
Size exclusion chromatography	Size	Poly(methacrylic acid-co-ethylene glycol dimethacrylate) [51]	Protein	Mixed of water and acetonitrile

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
