# Peer review of "Recent Developments of Liquid Chromatography Stationary Phases for Compound Separation: From Proteins to Small Organic Compounds"

_molecules, 2022, doi:10.3390/molecules27030907_

Round 1
Reviewer 1 Report
The review entitled 'Recent developments of liquid chromatography columns for molecule separation' by Handajaya Rusli et al. updates different LC techniques. The authors weighted the writing too much on affinity chromatography aspect/concept leading to lack of basic knowledge and information on other types of LC. For example, they considered Reverse-phase chromatography with the separation principal of affinity (Table 1). In general, affinity chromatography refers to specific binding interactions. Consideration of the Reverse-phase (without requirement of specific binding interactions) with the word 'affinity' can thus be unacceptable to many chromatographers. Several other critical concerns are listed below.
-The words 'molecule separation' are miss leading and not quite comprehensive for many references in this review which are mostly conventional LC separation. This should be replaced with more suitable words like 'compound separation'. Separation in LC conventionally refers to separation of peaks (one peak per one compound, mostly not one peak per one molecule). In general, one could write a peak of a compound (a compound consisting of many molecules) but not a peak of a molecule. An example against 'molecule separation' could be that conventional LC could separate compounds but could not separate molecules of the same compound. If the authors really want to use molecule separation, references about single or a few molecule separations (one peak per one molecule) or molecular level separation should be mostly discussed into this review.
-Several wordings in the Abstract are overclaimed and should be removed/replaced to be softer, such as
'regardless of the size of the target molecules' --> LC could not be used with solid/undissolved or highly volatile compounds OR the target molecules should be clearly defined here.
'unlimited combinations of mobile and stationary phases' --> Ones could combine anything but it should be effective. For example, chromatographers do not use pure aqueous with reverse phase separation. Also, the mobile phase should dissolve the analytes and should not destroy the stationary phase.
- Why the authors only focus on proteins, chiral molecules, and polycyclic aromatic hydrocarbons? There is no link between the aromatic hydrocarbons and the others. In fact, some other compounds such as ketones, phenolic compounds, pesticides and ions, are also mentioned in this review. It seems to me that the authors forgot updating the other compounds in the abstract.
-In the section of 2. The principle of separation of compounds in liquid chromatography, the authors only focuses on the interactions of affinity, electrostatic and size exclusion (e.g. Figure 1). This classification is very confusing and performed without proper citations. For example, affinity and electrostatic interactions can be overlapped. In addition, the conventional hydrophobic interaction employed in reverse-phase mode is missing. In fact, basic interactions in chromatography have been well described a decade ago, e.g. involving hydrophobicity, steric, hydrogen bond acidity, hydrogen bond basicity and electrostatic contributions [N.S. Wilson, M.D. Nelson, J.W. Dolan, L.R. Snyder, R.G. Wolcott, P.W. Carr, J.Chromatogr. A 961 (2002) 171–193]. Affinity interactions could be combination of such interactions plus site/shape specific interaction. The authors should present all of these interactions in Figure 1. Please also pay more attention to and discuss/update/rewrite more about these interaction types throughout the review.
-'Column', 'sample' and 'eluent' heads in Table 1 are better replaced with 'stationary phase', 'analyte' and 'mobile phase', respectively. Ionic chromatography and electrochromatography rows should be more carefully rewritten. For example, Nitric acid = pure acid?, Cr(III) and Cr(VI) = analysis of solid metal or ions?, pH buffer???
-Representation of ligand in Figure 2C with the helix shape should be avoided since this leads to the confusion that only the alpha helix ligand must be used.
- Figure 4 should be more clearly discussed what are the molecules and functions of the spacer and target molecules.
-What is the molecular structure of 'cam' in Figure 7 and the related discussions.
Author Response
"Please see the attachment."

Reviewer 2 Report
The Review in very interesting. However, I suggest a major revision before the publication.
Key points to be revised are:
- English language stile to improve the comprehension of the text
- Better discuss the advantages following the use of new stationary phases for the separation of the cited different molecules. In this form, the manuscript is a list of application without a proper discussion. Why should an analyst use these columns instead of others?
- Considering that proteins, chiral molecules and PAHs are the target molecules reported in this review, I also suggest to revise the title. In the present for it is too general.
- Revise typing errors
Author Response
"Please see the attachment."

Round 2
Reviewer 2 Report
Now, the paper is suitable for publication